# Improvement in the Hard Milling of AISI D2 Steel under the MQCL Condition Using Emulsion-Dispersed MoS₂ Nanosheets

**Pham Quang Dong ¹, Tran Minh Duc ¹, Ngo Minh Tuan ¹, Tran The Long ¹,\*, Dang Van Thanh ² and Nguyen Van Truong ³**

¹ Department of Manufacturing Engineering, Faculty of Mechanical Engineering, Thai Nguyen University of Technology, Thai Nguyen 250000, Vietnam; quangdongctm@tnut.edu.vn (P.Q.D.); minhduc@tnut.edu.vn (T.M.D.); minhtuanngo@tnut.edu.vn (N.M.T.)

² Faculty of Basic Sciences, College of Medicine and Pharmacy, Thai Nguyen University, Thai Nguyen 250000, Vietnam; thanhdv@tnmc.edu.vn

³ Faculty of Fundamental of Sciences, Thai Nguyen University of Technology, Thai Nguyen 250000, Vietnam; truonglyk3@gmail.com

\* Correspondence: tranthelong90@gmail.com or tranthelong@tnut.edu.vn; Tel.: +84-985-288-777

**Abstract:** The present work shows the process for MoS₂ nanosheet production by liquid N₂-quoched bulk, a novel method having highly efficient, green, and facile operation. The produced MoS₂ nanoparticles are suspended in minimum quantity cooling lubrication (MQCL)-based fluid to form nanofluid used for the hard milling of AISI D2 steel. The study aims to improve the hard-milling performance assisted by the MQCL technique using MoS₂ nanofluid. ANOVA analysis is used to evaluate the effects of three input machining variables, including nanoparticle concentration, cutting speed, and material hardness on cutting forces. The results indicate that the better cooling effect from the principle of the Ranque–Hilsch vortex tube of the MQCL device combined with the better lubricating performance from MoS₂ nanofluid brings out the sustainable alternative solution for machining difficult-to-cut material. Moreover, the experimental results provide the technical guides for the selection of proper values of nanoparticle concentration and cutting speed while ensuring the technological, economic, and environmental characteristics.

**Keywords:** liquid N₂-queched bulk; hard milling; MQCL; MoS₂ nanosheets; nanofluid; cutting force; difficult-to-cut material

## 1. Introduction

In the machining field, the manufacture of difficult-to-cut materials, especially hardened steel, has been growing concern not only for researchers but also for manufacturers around the world. Hardened steels have been widely used in different applications, such as mold and die, automotive components, bearings, and so on. They possess high hardness and strength properties, which makes them difficult to cut by using the conventional machining processes like turning, milling, and drilling due to large cutting forces, high cutting temperature, and severe tool wear. Traditionally, grinding operations have been the most common in machining hardened steel components (45–65 HRC). However, the low productivity, high cost, and environmental issues caused by cutting fluids are considered the main drawbacks of the grinding process [1]. Accordingly, the hard machining processes, such as hard turning, milling and drilling, have been developed to meet the growing demand of high productivity, good surface quality, coolant elimination, manufacturing cost reduction, and flexibility to different types of parts [2]. Hard turning was early recognized and applied in the automotive

industry, which proved the effectiveness in technological, economic, and environmental characteristics in machining the transmission components when compared to grinding. In addition, a typical example of early applications of hard turning is the surface of gear-wheel bearings by using polycrystalline cubic boron nitride (CBN) inserts. Nevertheless, the enormous cutting heat and large cutting forces are always the main problems when applying hard machining processes, which require the high-graded inserts like coated cemented carbide, ceramics, (Polycrystalline) Cubic Boron Nitride ((P)CBN), Polycrystalline Diamond (PCD) tools [3–6]. Furthermore, dry hard cutting is often used to avoid thermal shock leading to tool failure and retain the environmentally friendly character. This also limits the range of cutting conditions and the productivity as well as the increase of manufacturing cost. Therefore, the alternative solutions have been studied and proposed to improve the cutting performance of hard machining processes, including non-traditional cutting processes assisted in machining, such as laser-assisted machining (LAM) [7,8], minimum quantity lubrication (MQL) and minimum quantity cooling lubrication (MQCL) assisted in machining and nanofluids enriched in MQL and MQCL-assisted machining [9–17]. Among these novel approaches, the utilization of nano additives suspended in MQL- and MQCL-based fluids was proven the significant improvement in hard cutting performance. B. Li and his co-authors (2017) [18] investigated the performance of the heat transfer of different types of nanoparticles ($MoS_2$, $ZrO_2$, CNT, PCD, $Al_2O_3$, $SiO_2$) enriched in MQL-based fluid in the grinding process of Ni-based alloy. The authors pointed out that the viscosity and thermal conductivity of MQL-based fluid improved by suspending nanoparticles, from which grinding forces and heat significantly decreased. F. Pashmforoush et al. (2018) [19] also made the study on the grinding process of Inconel 738 super alloy by using dry condition, conventional fluid and CuO nanofluid. The values of the wheel loading ratio and surface roughness decreased when compared to dry grinding and conventional fluid grinding. The nanoparticles' concentration and feed velocity have strong effects on surface roughness and wheel loading. Pil-Ho Lee et al. (2012) [20] studied the diamond and $Al_2O_3$ NFMQL (nanofluid minimum quantity lubrication) performance in micro grinding. The authors concluded that the nano additives contribute to reduce grinding forces significantly and enhance the surface quality, especially in the case of using higher nanoparticles concentration and smaller grain size. Y. Zhang et al. (2016) [21] had done the study on mass volume of nanoparticles $MoS_2$ and CNTs (carbon nanotubes) in MQL-based fluid in grinding process of Ni-based alloy. The lubrication performance improved and contributed to the better surface quality. Furthermore, the authors also investigated hybrid $MoS_2$-CNTs nanofluid and made the comparison to the nanofluid with only one nanoparticle type. The obtained results indicated that the surface quality is better by using hybrid $MoS_2$-CNTs nanofluid. On the other hand, there have been many studies on the NFMQL machining processes of difficult-to-cut materials using geometrical-defined cutting edges like turning, milling, and drilling. Ç. V. Yıldırım et al. (2019) [22] studied the effect of hBN nanofluid-MQL on tool wear, tool life, surface roughness and cutting temperature in the turning of Ni-based Inconel 625. The obtained results indicated that the tool life was considerably prolonged under NFMQL with 0.5 vol% hBN when compared to dry and pure MQL machining. The main reason for this is that the presence of hBN nanoparticles in oil mist enhances the cooling and lubricating effects, so the cutting forces and cutting temperature reduce, which decreases tool wear. In addition, the hBN ratio in coolant is a very important factor, which strongly influences the machining performance. In 2020, the author newly made the investigation of Cryogenic cooling by liquid nitrogen and NFMQL used in the hard turning of AISI 420 [23]. He concluded that the surface quality under NFMQL is better, while machining outputs in term of cutting temperature, tool life, tool wear, and chip morphology under cryogenic cooling show better results. The main reason for this observation is the superior cooling performance of liquid nitrogen compared to NFMQL, especially at high cutting speed. Another promising approach is the use of minimum quantity cooling lubrication (MQCL), which is the development of the MQL method to improve the cooling effect. This technique has been gaining growing attention for machining difficult-to-cut materials. S. Pervaiz et al. (2017) [24] investigated the turning process of Ti alloy under MQCL condition compared to dry and flood cutting. The better cooling and lubricating performance

is reported and contributes to reducing the friction coefficient in cutting zone, leading to the decrease in cutting forces and tool wear as well as the enhancement of surface quality. R. W. Maruda et al. (2017) [25] had done the investigation on hard turning of AISI 1045 steel under MQCL condition with emulsion fluid. The results revealed that the oil mist formed in cutting zone contributes to improve cooling and lubricating effects. The authors also investigated the formation of droplets and the relationship between the MQCL parameters and diameter [26]. In another study, they concluded that the chip shape is favorable and the decrease of the chip thickening coefficient is reported under MQCL condition due to better cooling and lubricating effectiveness [27]. The quadratic effects of MQCL parameters and droplets are also studied to bring out the possibility to choose the suitable condition for oil mist formation on the newly machined surface within 1 s and then evaporate [28]. However, the cooling effect of the MQCL methods used by the mentioned studies is based on the cutting fluid like emulsion, which already possesses the cooling and lubricating properties. The application of the MQCL technique with the real cooling effect is definitely the novel approach to improve the machining performance and enlarge MQCL applicability as well as to bring out the new alternative solutions for difficult-to-cut materials. Furthermore, MQCL using nanofluid is an up-to-date topic and is needed to study [17,29].

Among the common types of nanoparticles, molybdenum disulfide ($MoS_2$) possesses a low friction coefficient and excellent physical properties. This kind of nanoparticle also has a large active surface area, high reactivity, and increased adsorption capacity compared to the bulk material. Hence, $MoS_2$ nanofluid is found to be advantageous for improving the lubricating effect for machining processes. Several approaches have been proposed for producing $MoS_2$ nanosheet, including liquid exfoliation [30], chemical vapor deposition [31], thermal decomposition [32], the sulfurization of an Mo-based compound [33] and vapor–solid growth [34]. For instance, Zhang et al. [30] reported a fast and highly controllable approach to yield a variety of semiconducting nanosheets by exfoliation. Taking the benefit of the electrochemical lithiation discharge process, bulk $MoS_2$ could be intercalated by lithium ion and then a high-yield (92%), single-layer $MoS_2$ was achieved after subsequent ultrasonication. Ku et al. [32] successfully prepared a bi- and tri-layer of $MoS_2$ nanosheets by simple thermolysis of $(NH_4)_2SO_4$ at specific temperature. D.V. Thanh and our group [35] presented a novel and facile method by the exfoliation of liquid $N_2$–quenched bulk $MoS_2$. This technique is promising for further application owning to its highly efficient, green, and facile operation. The produced $MoS_2$ nanoparticles are suspended in MQCL-based fluid to form $MoS_2$ nanofluid used for the hard milling of AISI D2 steel. This paper aims to improve the cutting performance of the hard-milling process by applying novel cooling and lubricating method, nanofluid minimum quantity cooling lubrication (NFMQCL). The results indicate the promising alternative solution for machining difficult-to-cut materials while retaining the environmentally friendly characteristics, suitable for sustainable production.

## 2. Materials and Methods

### 2.1. The Production of $MoS_2$ Nanoparticles

The preparation of $MoS_2$ nanosheets by liquid $N_2$-queched bulk $MoS_2$ was reported in [35]. Briefly, 500 mg of $MoS_2$ bulk was transferred to 100 mL of 5% KOH solution under vigorous stirring for 30 min. The mixture was maintained at 80 °C for 24 h and then it was quickly quenched in liquid $N_2$. The temperature dramatically dropped from 80 to −196 °C when the $MoS_2$ solution was immersed in liquid $N_2$. This prompt temperature decreasing not only expands the $MoS_2$ but also allows for the intercalation of both $K^+$ ions and $N_2$ gas into the interlayer, resulting in exfoliating $MoS_2$ nanosheets under the sonication synergistic. After quenching, the dispersion was sonicated at 20 kHz under a power of 400 W. The resulting material was collected through vacuum filtration with PVDF membranes (average pore size: 0.2 μm). The final product was dried in 50 °C under vacuum for 24 h. The Raman spectra of bulk $MoS_2$ and $MoS_2$ nanosheets powders were recorded using a Raman Spectrometer JobinYvon LabRAM HR800 (FOCAS Research Institute, Dublin, Ireland).

Transmission electron microscopy (TEM) images were recorded using a JEM-2100F Field Emission Electron Microscope (JEOL Ltd., Tokyo, Japan).

Figure 1 displays the transmission electron microscopy (TEM) image, scanning electron microscopy (SEM) image, lateral size distribution and Raman spectra of $MoS_2$ flake obtained from the $N_2$-quenched exfoliation method [35]. A transparent to the electron beam corresponding to the sheet-like structures of $MoS_2$ flakes with a lateral size of approximately several hundred nanometers was shown in Figure 1a and the thickness ranges from 10 to 20 nm. Figure 1b displays the SEM image of $MoS_2$ flakes which clearly show the sheets morphology with the lateral size from several hundred nm to approximate 2 μm. Furthermore, a statistic was estimated from 75 objects in SEM images to evaluate the lateral size distribution of $MoS_2$ nanosheets as shown in Figure 1c. $MoS_2$ nanosheets almost reveal the lateral size smaller than 1 μm, which is consistent with the TEM data. In addition, the Raman spectra in Figure 1d reveal a typical of in-plane $E^1_{2g}$ and out-plane $A_{1g}$ Raman peaks of $MoS_2$ flakes located at 376 and 403.2 $cm^{-1}$, respectively, which highly agreed with the previous reports [36,37].

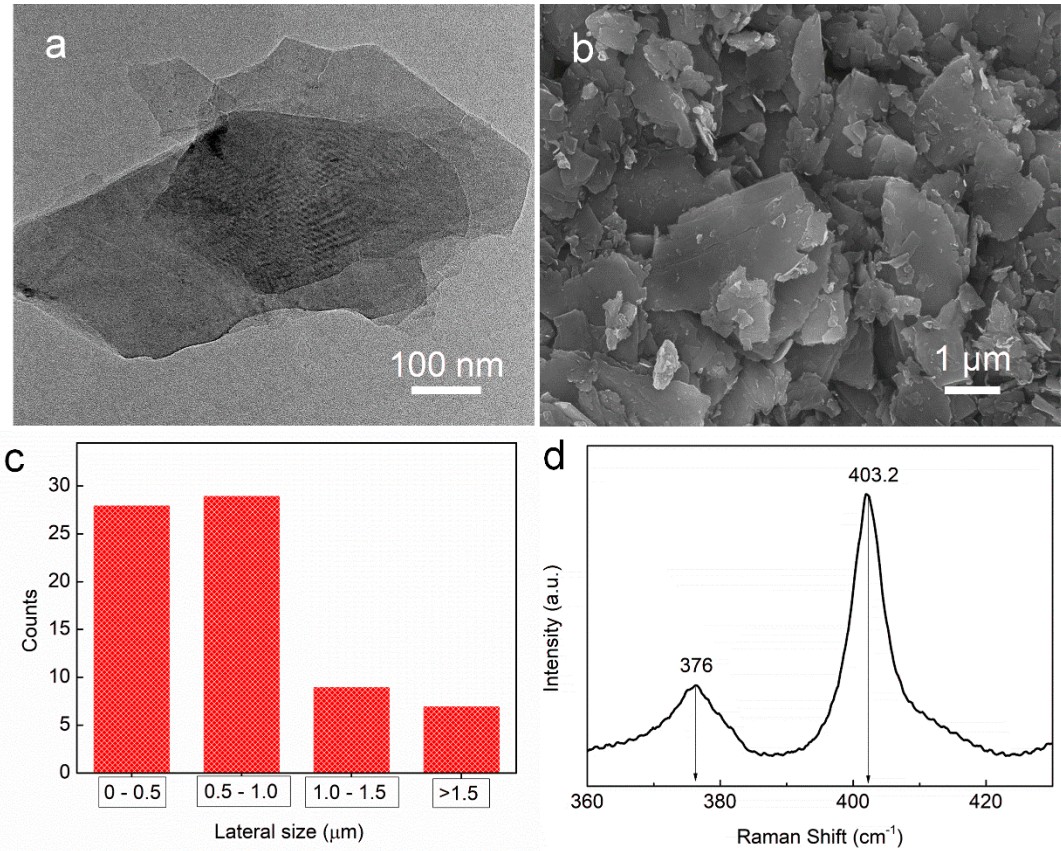

**Figure 1.** (**a**) Transmission electron microscopy (TEM) image, (**b**) scanning electron microscopy (SEM) image, (**c**) Lateral size distribution and Raman spectra (**d**) of $MoS_2$ nanosheets.

## 2.2. Experimental Set Up

The experimental devices consist of Mazak vertical center smart 530C (Yamazaki Mazak Corporation, Aichi, Japan), tool holder type with the designation SHIJIE BAP 400R-50-22-4T with the diameter of 50 mm, and APMT 1604 PDTR LT30 PVD submicron carbide inserts (LAMINA TECHNOLOGIES SA, Yverdons-les-Bains, Switzerland) with a flank angle of 11°, nose radius of 0.66 mm, and a TiAlN coating layer. The samples are AISI D2 steels (52 ÷ 60 HRC) with the dimensions of 90 mm × 48 mm × 50 mm and their chemical composition shown in Table 1.

**Table 1.** Chemical composition of AISI D2 steel (According to American Society for Testing and Materials (ASTM) A681).

| Chemical Composition (%) | | | | | | | | | |
|---|---|---|---|---|---|---|---|---|---|
| C | Si | Mn | Ni | Cr | Mo | W | V | P | S |
| 1.4–1.6 | 0.1–0.6 | 0.1–0.6 | 0.5 | 11.0–13.0 | 0.7–1.2 | 0.2–0.5 | 0.5–1.1 | 0.03 | 0.03 |

The MQCL system includes Frigid-X Sub-Zero Vortex Tool Cooling Mist System (Nex Flow™ Air Products Corp, Richmond Hill, ON, Canada) compressed air, pressure stabilization device, water-based emulsion 5% and $MoS_2$ nanoparticles. The experiment set up is shown in Figure 2.

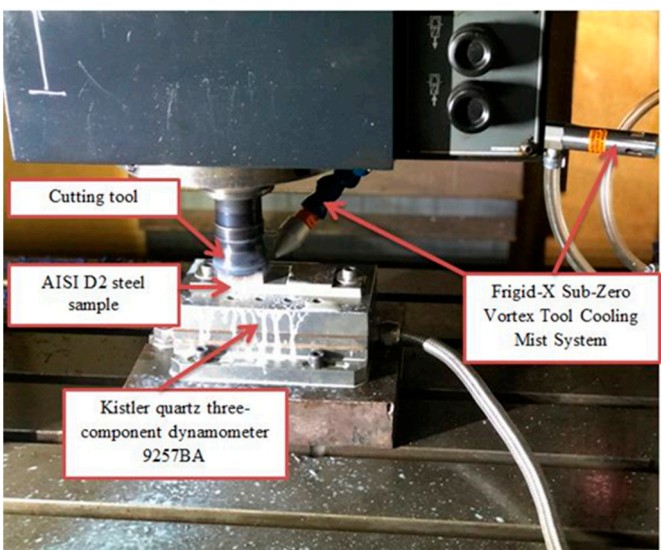

**Figure 2.** Experimental set up.

To ensure the uniform suspension of $MoS_2$ nanosheets in emulsion-based fluids, the prepared nanofluids are kept in Ultrasons-HD ultrasonicator (JP Selecta, Abrera (Barcelona), Spain), generating 400 W ultrasonic pulses at 40 kHz for 2 h to ensure the uniform distribution of the nanoparticles. In order to use the obtained nanofluids effectively and avoid the precipitation of agglomerated nanoparticles during the long time of machining, the nanofluid was placed in the 3000868-Ultrasons-HD and directly used for the MQCL system.

Measuring equipment consists of Kistler quartz three-component dynamometer 9257BA (Kistler Instruments (Pte) Ltd., Midview, Singapore).

*2.3. Experiment Design*

The fixed parameters for cutting condition include the feed rate of 0.012 mm/tooth, a depth of cut of 0.12 mm, air pressure of 0.6 MPa, flow rate of 30 mL/h; the room temperature 24 ÷ 27 °C; the temperature of output cool air through MQCL nozzle of 4 ÷ 8 °C. Box–Behnken experimental design is made with the help of Minitab 18.0 software (N03-T5 Embassy Garden, Ha Noi, Vietnam) (Table 2). Table A1 summarizes the design of experiment with test run order and output in term of cutting forces $F_x$, $F_y$, $F_z$. Each test run is repeated by three times under the same cutting parameters.

**Table 2.** Control factors and their levels.

| Control Factor | Unit | Symbol | Level | |
|---|---|---|---|---|
| | | | Low | High |
| Nanoparticle concentration (*np*) | wt.% | $x_1$ | 0.5 | 1.5 |
| Cutting speed (*V*) | m/min | $x_2$ | 90 | 110 |
| Hardness | HRC | $x_3$ | 52 | 60 |

## 3. Results and Discussion

*The Effects of Input Machining Parameters on Cutting Forces*

The ANOVA analysis is carried out at a confidence level of 95% (i.e., 5% significance level). Tables A2–A4 show the results of the ANOVA analysis. The regression models of cutting forces $F_x$, $F_y$, $F_z$ with $R^2$ equal to 89.96, 94.21, 98.17, respectively, are given below in Equations (1)–(3).

$$F_x = -1422 - 18.9X_1 + 4.52X_2 + 44.1X_3 + 9.48X_1 * X_1 - 0.0232X_2 * X_2 - 0.383X_3 * X_3 \tag{1}$$

$$F_y = -2248 - 160.4X_1 + 25.03X_2 + 133.1X_3 + 71.7X_1 * X_1 - 0.1232X_2 * X_2 - 1.198X_3 * X_3 \tag{2}$$

$$F_z = -2403 - 158.0X_1 + 10.0X_2 + 73.2X_3 + 70.6X_1 * X_1 - 0.0519X_2 * X_2 - 0.621X_3 * X_3 \tag{3}$$

The Pareto charts of the standardized effects with $\alpha = 0.05$ for the response parameters $F_x$, $F_y$, $F_z$ are shown in Figures 3–5. Hardness ($x_3$) has the strongest influence on $F_x$, $F_z$, while nanoparticle concentration ($x_1$) has a strongest influence on $F_y$ and also causes strong effect on $F_z$. It can be seen that cutting speed ($x_2$) has a very little effect on cutting forces $F_x$, $F_y$, $F_z$.

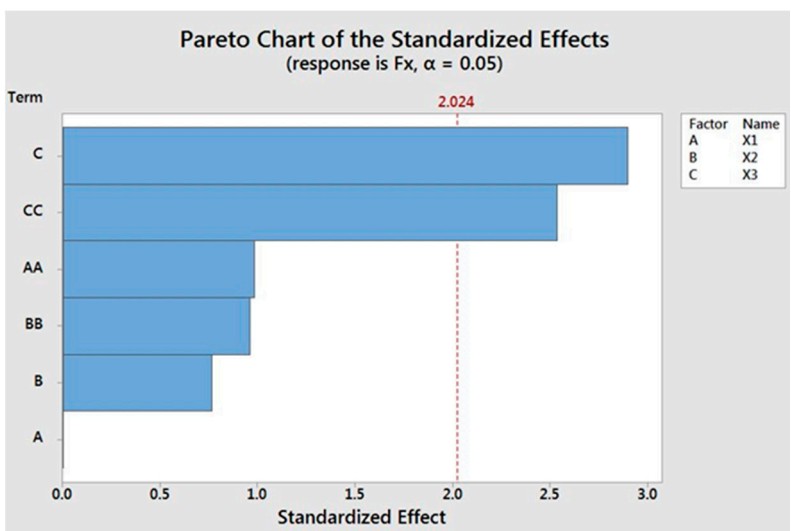

**Figure 3.** Pareto chart of effects of investigated factors on cutting force $F_x$. (A is $x_1$: nanoparticle concentration, B is $x_2$: cutting speed, C is $x_3$: hardness, AA is the quadratic effect of nanoparticle concentration, BB is the quadratic effect of cutting speed, and CC is the quadratic effect of hardness).

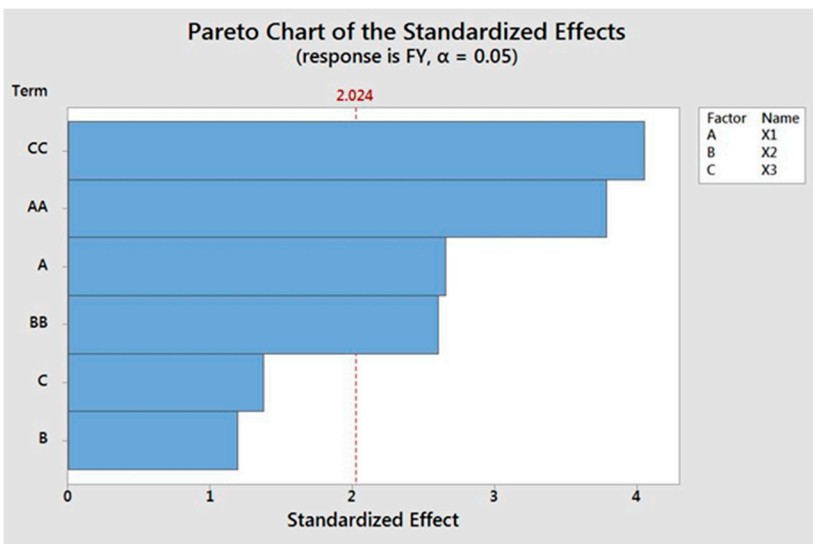

**Figure 4.** Pareto chart of effects of investigated factors on cutting force $F_y$. (A is $x_1$: nanoparticle concentration, B is $x_2$: cutting speed, C is $x_3$: hardness, AA is the quadratic effect of nanoparticle concentration, BB is the quadratic effect of cutting speed, and CC is the quadratic effect of hardness).

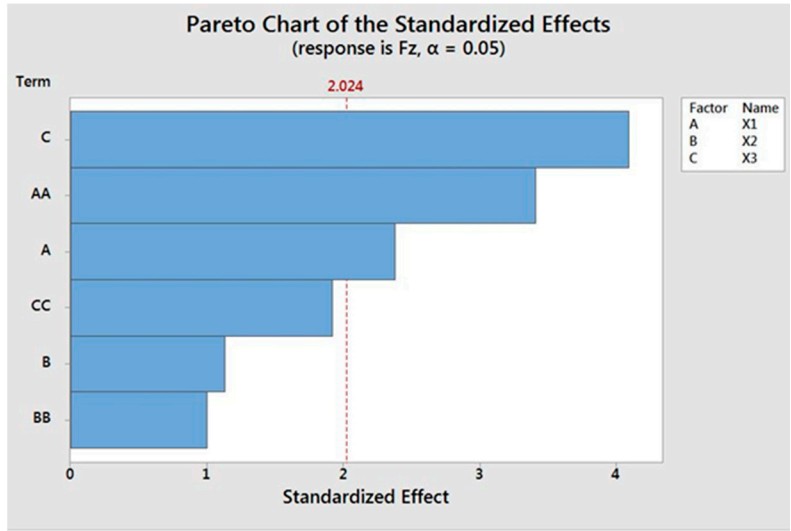

**Figure 5.** Pareto chart of effects of investigated factors on cutting force $F_z$. (A is $x_1$: nanoparticle concentration, B is $x_2$: cutting speed, C is $x_3$: hardness, AA is the quadratic effect of nanoparticle concentration, BB is the quadratic effect of cutting speed, and CC is the quadratic effect of hardness).

The quadratic effect CC ($x_3x_3$) reveals the significant influence on the investigated functions of $F_x$, $F_y$, and the quadratic effect AA ($x_1x_1$) exhibits the significant effect on the investigated functions of $F_y$, $F_z$. In addition, the quadratic effect BB ($x_2x_2$) has a strong influence on $F_y$. The other quadratic effects of $x_1x_2$, $x_1x_3$, $x_2x_3$ cause very little influences and are not investigated in the regression models. Studying the charts in Figures 3–5, the proper selection of nanoparticle concentration and hardness is needed to study for the effects on cutting forces in order to improve the machining performance while retaining the good tool life and surface quality as well.

The surface and contour plots of investigated variables on cutting force $F_x$ are shown in Figures 6–8.

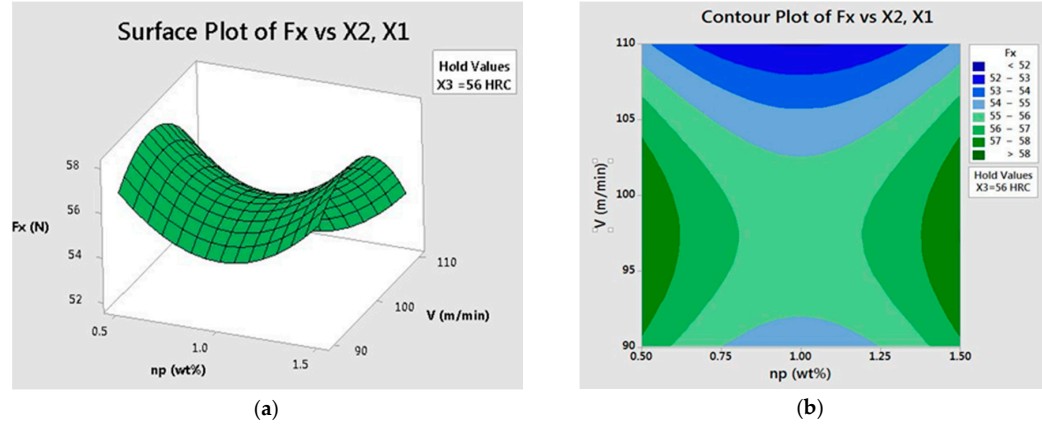

**Figure 6.** Effects of nanoparticle concentration and cutting speed on cutting force $F_x$: (**a**) surface plot, (**b**) contour plot.

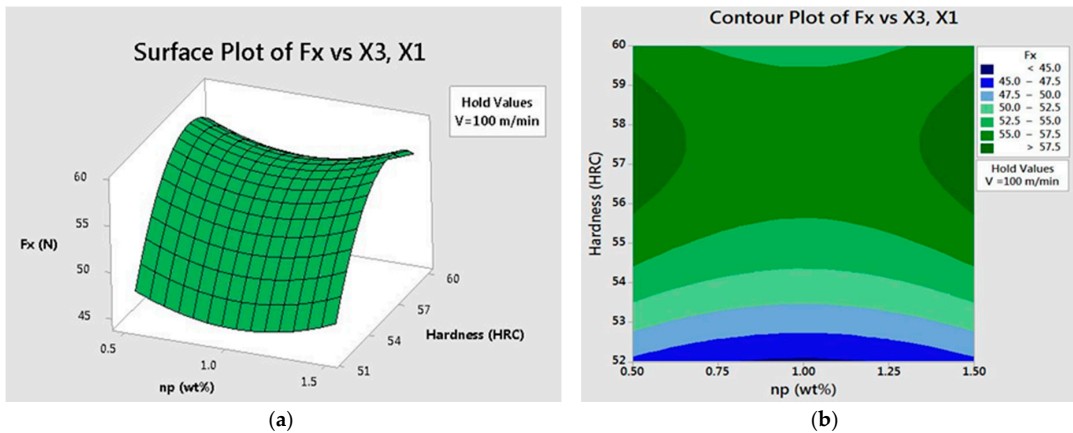

**Figure 7.** Effects of nanoparticle concentration and hardness on cutting force $F_x$: (**a**) surface plot, (**b**) contour plot.

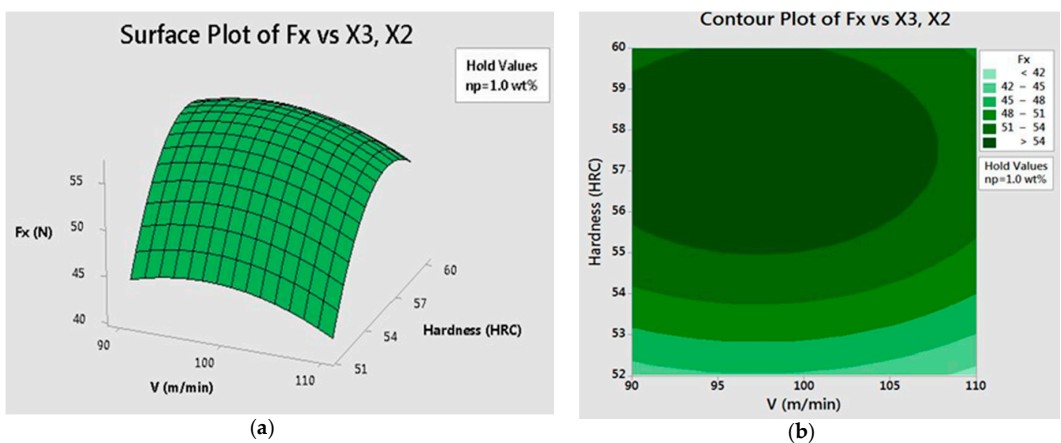

**Figure 8.** Effects of cutting speed and hardness on cutting force $F_x$: (**a**) surface plot, (**b**) contour plot.

The graphs of effects of nanoparticle concentration and cutting speed with constant hardness of 56 HRC are shown in Figure 6. From Figure 6a, the cutting force $F_x$ increases with the rise of cutting speed from 90 to 100 m/min and then decreases when cutting speed rises to 110 m/min, but the surface plot is contrast with the changes of nanoparticle concentration. Observed from Figure 6b, the aim for low cutting force $F_x$ can achieve by using high level of cutting speed (110 m/min) and np of 1.0 wt.%. When holding cutting speed of 100 m/min or nanoparticle concentration of 1.0 wt.%, it can be clearly

seen that the hardness factor has a very strong effect on cutting forces $F_x$ (Figures 7 and 8). The force $F_x$ rapidly goes up with the rise of material hardness.

The surface and contour plots of nanoparticle concentration and cutting speed on cutting forces $F_y$, $F_z$ while keeping the constant hardness of 56HRC are shown in Figures 9 and 10.

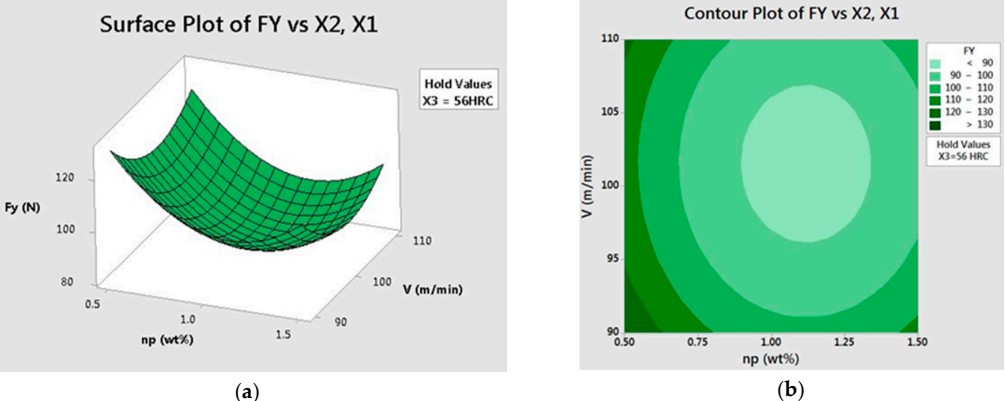

(**a**) 　　　　　　　　　　　　　　　　　　　　　　　　　　　　(**b**)

**Figure 9.** Effects of nanoparticle concentration and cutting speed on cutting force $F_y$: (**a**) surface plot, (**b**) contour plot.

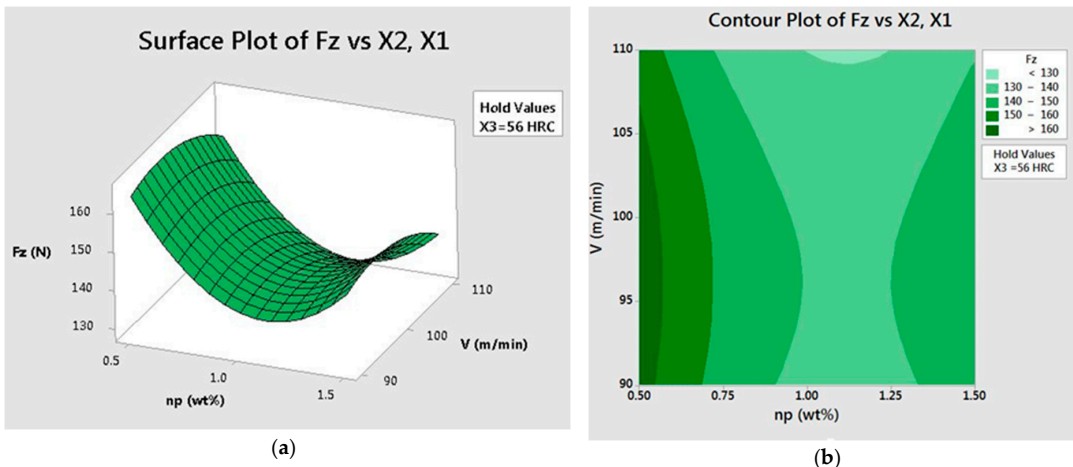

(**a**) 　　　　　　　　　　　　　　　　　　　　　　　　　　　　(**b**)

**Figure 10.** Effects of nanoparticle concentration and cutting speed on cutting force $F_z$: (**a**) surface plot, (**b**) contour plot.

From the surface and contour plots in Figures 9 and 10, it clearly reveals that the value of nanoparticle concentration of 1.0 wt.% brings out the decrease of cutting forces $F_y$, $F_z$. For smallest values of $F_y$, cutting speed can be chosen from 95 to 105 m/min (Figure 9b), and for smallest values of $F_z$, it can range from 90 to 110 m/min (Figure 10b). Hence, depending on the specific conditions, these results could provide the useful guide for improving hard milling performance while ensuring the technical and economic characteristics.

The surface and contour plots of hardness and nanoparticle concentration on cutting forces $F_y$, $F_z$ while cutting speed is constant with V = 100 m/min are shown in Figures 11 and 12.

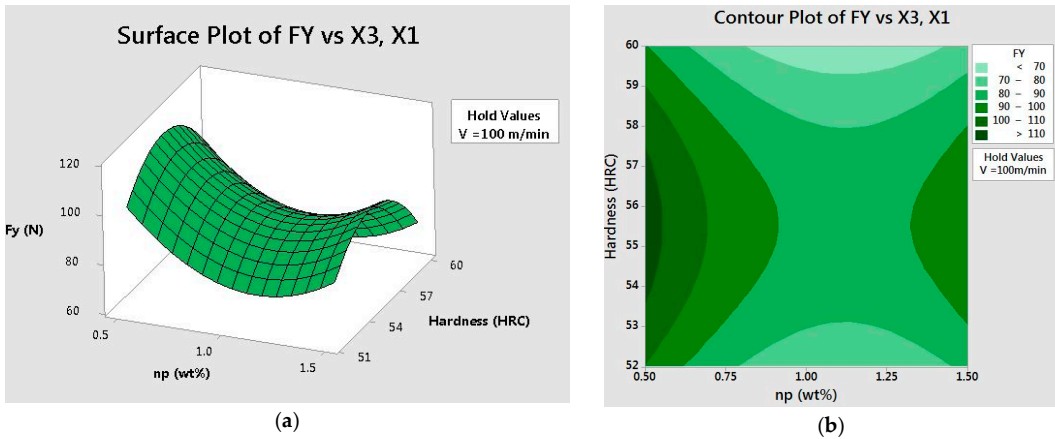

**Figure 11.** Effects of nanoparticle concentration and hardness on cutting force $F_y$: (**a**) surface plot, (**b**) contour plot.

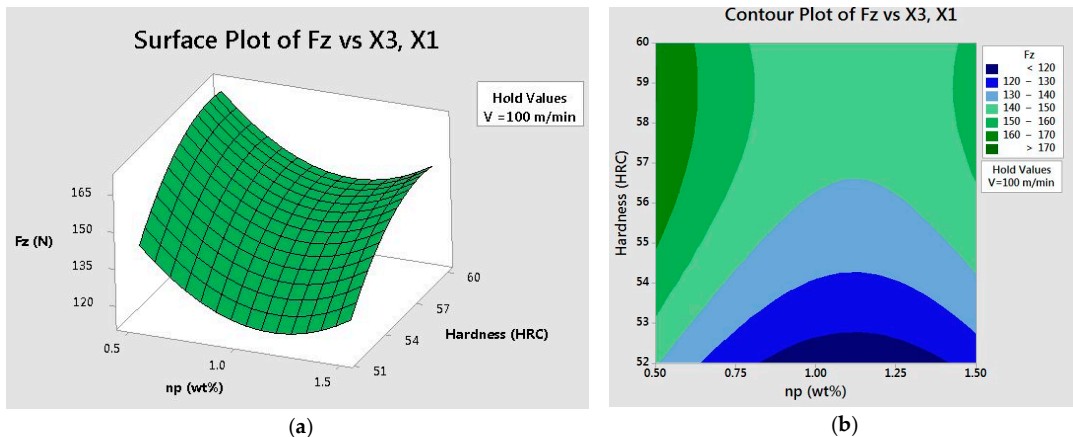

**Figure 12.** Effects of nanoparticle concentration and hardness on cutting force $F_z$: (**a**) surface plot, (**b**) contour plot.

From the surface and contour plots in Figures 11 and 12, it clearly reveals that the value of nanoparticle concentration about 1.0 wt.% also gives out the decrease of cutting forces $F_y$, $F_z$. The cutting forces $F_y$, $F_z$ grow with the increase of material hardness from 52 to 56 HRC, but there is a little difference in the hardness range of 56–60 HRC. The force $F_y$ exhibits the reducing trend while the force $F_z$ increases.

The surface and contour plots of hardness and cutting speed on cutting forces $F_y$, $F_z$ while nanoparticle concentration is constant with np = 1.0 wt.% are shown in Figures 13 and 14. In general, when cutting speed increases, the forces $F_y$, $F_z$ decrease but not much. From these results, nanoparticle concentration can be chosen by about 1.0 wt.% in combination with a low hardness value, and a high cutting speed can be used for increasing productivity.

In detail, $F_y$ is the thrust force, the so-called springback of the work materials, which plays a significant role in hard machining and has a strong effect dimensional accuracy and flank wear. Moreover, it causes high contact stresses over the tool flank contact face and this is a distinguishing feature of hard machining. $F_z$ is the main cutting force, which has a strong effect on the cutting performance and productivity. From the study, the technical guides will be given for selecting the cutting condition and making further studies.

In the machining of hardened steels with geometrically defined cutting tools, catastrophic failure causes the saw-tooth chip formation in the primary shear zone. Thermal stresses originate mainly in the intensive friction between the flank face and machined face. Moreover, the plastic deformation is very little, so the thermal stresses and intensive friction cause white layer [2]. Therefore, the presence

of MoS$_2$ nanosheet having a low friction coefficient and excellent lubricating property in MQCL-based fluid contributes to reducing the friction stresses in contact zones [38,39], from which the cutting forces decrease significantly and the machining performance improves.

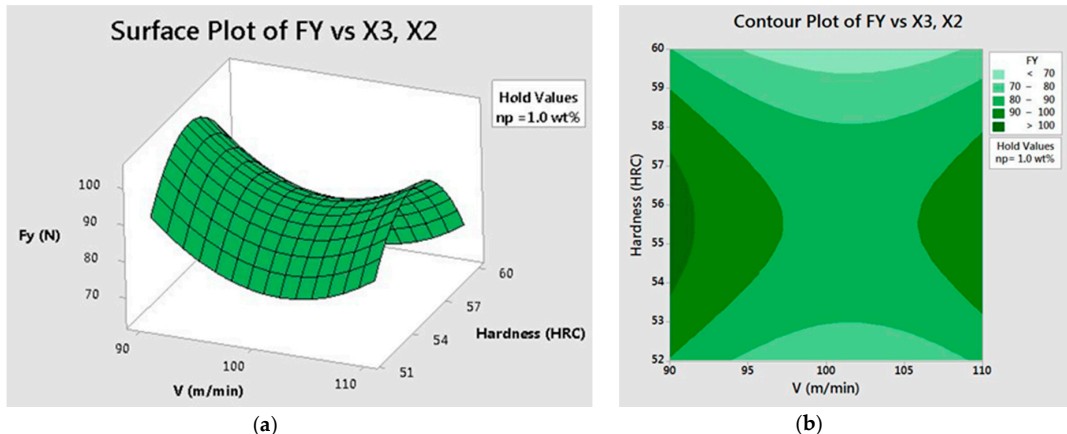

**Figure 13.** Effects of cutting speed and hardness on cutting force F$_y$: (**a**) surface plot, (**b**) contour plot.

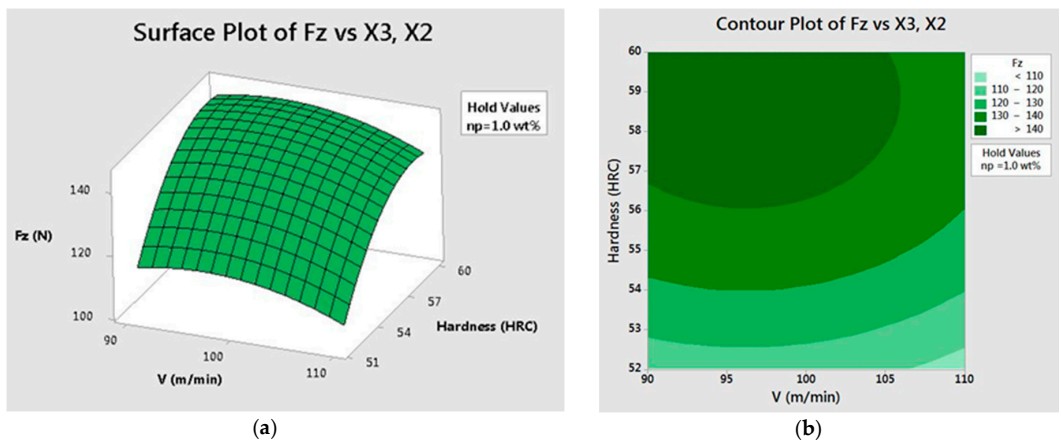

**Figure 14.** Effects of cutting speed and hardness on cutting force F$_z$: (**a**) surface plot, (**b**) contour plot.

## 4. Conclusions

In this study, MoS$_2$ nanoparticles are produced by exfoliation of liquid N$_2$–quenched bulk, which is the facile method owning to its highly efficient, green, and facile operation. The produced MoS$_2$ nanosheets are directly used for the MQCL hard milling process. Box–Behnken experimental design with the help of ANOVA analysis is used to investigate the effects of three variables, including nanoparticle concentration, cutting speed, and material hardness on cutting force components F$_x$, F$_y$, F$_z$.

From the obtained results, hardness and nanoparticle concentration reveal the strongest influence on the cutting force functions. The nanoparticle concentration should be used with 1.0 wt.% for reducing cutting forces in combination with a high cutting speed of 110 m/min in order to achieve the technological and economic characteristics. The cutting force components grow up with the increase of hardness.

The machining performance of hard milling process improves under the MQCL condition by using an MQCL device named the Frigid-X Sub-Zero Vortex Tool Cooling Mist System with a real cooling effect based on the principle of the Ranque–Hilsch vortex tube. The cooling and lubricating effects significantly enhance due to the use of the novel MQCL device and MoS$_2$ nanofluid with excellent lubricating property. Therefore, the normal coated cemented carbide inserts, such as APMT 1604 PDTR LT30, could be effectively used for the hard machining of AISI D2 (52–60 HRC) with a high cutting

speed of 110 m/min, which is about 2.0–2.2 times higher than the manufacturer's recommendations [40]. This will contribute to enlarge the cutting applicability and decrease the manufacturing cost.

Moreover, this is one of the first studies of this type of MQCL tool using nanofluid for difficult-to-cut materials. Compared to our previous study, which concentrated on the surface roughness, surface microstructure, and surface profile and used the commercial $MoS_2$ nanoparticles [17], this work is a big development because $MoS_2$ nanoparticles are produced by our own method and cutting forces $F_x$, $F_y$, $F_z$ are investigated in detail, from which the technical guides are provided and further work can be made.

Furthermore, the MQCL tool only needs ordinary air rather than cryogenic $CO_2$ or $N_2$ for cooling, which will simplify the equipment and reduce the cost.

In further research, deeper studies are needed to focus on the effects of $MoS_2$ nanoparticles on tool wear and tool life as well as the lubricating behavior in the cutting zone. In addition to this, the morphology, size and production methods of $MoS_2$ nanosheets will be investigated.

**Author Contributions:** Conceptualization, P.Q.D., T.M.D., T.T.L. and D.V.T.; Data curation, N.M.T.; Formal analysis, P.Q.D., T.T.L., D.V.T., N.M.T. and N.V.T.; Investigation, P.Q.D., T.T.L., N.M.T. and N.V.T.; Methodology, P.Q.D., T.T.L., D.V.T., N.M.T. and N.V.T.; Project administration, T.M.D.; Resources, D.V.T.; Software, T.T.L., D.V.T. and N.V.T.; Supervision, N.V.T.; Validation, N.M.T.; Visualization, T.T.L.; Writing—original draft, P.Q.D., T.M.D., T.T.L. and D.V.T.; Writing—review & editing, P.Q.D., T.M.D., T.T.L. and D.V.T. All authors have read and agreed to the published version of the manuscript.

**Funding:** This research was funded by Vietnam Ministry of Education and Training with the project number of B2019-TNA-02.

**Acknowledgments:** The study had the support of Vietnam Ministry of Education and Training and Thai Nguyen University of Technology, Thai Nguyen University with the project number of B2019-TNA-02.

**Conflicts of Interest:** The authors declare no conflict of interest.

## Appendix A

**Table A1.** Design of the experiment with test run order and output in terms of cutting forces.

| Std Order | Run Order | PtType | Blocks | Input Machining Parameters | | | Response Variables | | |
|---|---|---|---|---|---|---|---|---|---|
| | | | | $x_1$ (wt.%) | $x_2$ (m/min) | $x_3$ (HRC) | $F_x$ (N) | $F_y$ (N) | $F_z$ (N) |
| 1 | 20 | 2 | 1 | 0.5 | 90 | 56 | 60.50 | 116.70 | 179.90 |
| 2 | 30 | 2 | 1 | 1.5 | 90 | 56 | 62.70 | 101.30 | 135.70 |
| 3 | 7 | 2 | 1 | 0.5 | 110 | 56 | 43.70 | 99.80 | 131.83 |
| 4 | 41 | 2 | 1 | 1.5 | 110 | 56 | 53.10 | 90.08 | 136.40 |
| 5 | 4 | 2 | 1 | 0.5 | 100 | 52 | 45.40 | 125.24 | 157.80 |
| 6 | 39 | 2 | 1 | 1.5 | 100 | 52 | 47.90 | 103.10 | 129.80 |
| 7 | 32 | 2 | 1 | 0.5 | 100 | 60 | 52.60 | 73.20 | 120.48 |
| 8 | 14 | 2 | 1 | 1.5 | 100 | 60 | 45.77 | 69.90 | 137.90 |
| 9 | 16 | 2 | 1 | 1 | 90 | 52 | 41.40 | 90.70 | 98.37 |
| 10 | 29 | 2 | 1 | 1 | 110 | 52 | 37.45 | 82.60 | 97.70 |
| 11 | 22 | 2 | 1 | 1 | 90 | 60 | 72.20 | 86.30 | 168.90 |
| 12 | 18 | 2 | 1 | 1 | 110 | 60 | 45.04 | 76.90 | 149.53 |
| 13 | 23 | 0 | 1 | 1 | 100 | 56 | 57.40 | 90.10 | 140.85 |
| 14 | 43 | 0 | 1 | 1 | 100 | 56 | 50.00 | 77.90 | 134.94 |
| 15 | 19 | 0 | 1 | 1 | 100 | 56 | 53.70 | 83.95 | 137.90 |
| 16 | 11 | 2 | 1 | 0.5 | 90 | 56 | 61.50 | 170.10 | 178.70 |
| 17 | 21 | 2 | 1 | 1.5 | 90 | 56 | 50.40 | 103.10 | 136.00 |
| 18 | 13 | 2 | 1 | 0.5 | 110 | 56 | 55.40 | 120.96 | 173.52 |
| 19 | 37 | 2 | 1 | 1.5 | 110 | 56 | 52.20 | 100.20 | 142.79 |
| 20 | 38 | 2 | 1 | 0.5 | 100 | 52 | 42.80 | 79.70 | 151.65 |

**Table A1.** *Cont.*

| Std Order | Run Order | PtType | Blocks | Input Machining Parameters | | | Response Variables | | |
|---|---|---|---|---|---|---|---|---|---|
| | | | | $x_1$ (wt.%) | $x_2$ (m/min) | $x_3$ (HRC) | $F_x$ (N) | $F_y$ (N) | $F_z$ (N) |
| 21 | 33 | 2 | 1 | 1.5 | 100 | 52 | 52.40 | 105.10 | 145.79 |
| 22 | 12 | 2 | 1 | 0.5 | 100 | 60 | 54.30 | 80.45 | 154.08 |
| 23 | 9 | 2 | 1 | 1.5 | 100 | 60 | 62.59 | 87.80 | 171.50 |
| 24 | 28 | 2 | 1 | 1 | 90 | 52 | 43.10 | 83.96 | 92.85 |
| 25 | 45 | 2 | 1 | 1 | 110 | 52 | 47.60 | 99.50 | 112.40 |
| 26 | 10 | 2 | 1 | 1 | 90 | 60 | 40.40 | 68.53 | 141.90 |
| 27 | 2 | 2 | 1 | 1 | 110 | 60 | 37.30 | 80.83 | 134.87 |
| 28 | 15 | 0 | 1 | 1 | 100 | 56 | 65.70 | 94.90 | 145.03 |
| 29 | 36 | 0 | 1 | 1 | 100 | 56 | 50.40 | 84.60 | 130.40 |
| 30 | 27 | 0 | 1 | 1 | 100 | 56 | 58.05 | 89.75 | 137.50 |
| 31 | 6 | 2 | 1 | 0.5 | 90 | 56 | 46.20 | 151.90 | 160.10 |
| 32 | 17 | 2 | 1 | 1.5 | 90 | 56 | 58.60 | 102.50 | 138.70 |
| 33 | 5 | 2 | 1 | 0.5 | 110 | 56 | 58.60 | 124.90 | 163.76 |
| 34 | 3 | 2 | 1 | 1.5 | 110 | 56 | 56.40 | 114.70 | 140.28 |
| 35 | 35 | 2 | 1 | 0.5 | 100 | 52 | 43.50 | 65.80 | 112.00 |
| 36 | 25 | 2 | 1 | 1.5 | 100 | 52 | 52.60 | 77.00 | 141.55 |
| 37 | 42 | 2 | 1 | 0.5 | 100 | 60 | 61.90 | 90.95 | 145.90 |
| 38 | 1 | 2 | 1 | 1.5 | 100 | 60 | 48.90 | 70.10 | 132.25 |
| 39 | 31 | 2 | 1 | 1 | 90 | 52 | 40.77 | 87.90 | 126.20 |
| 40 | 26 | 2 | 1 | 1 | 110 | 52 | 40.50 | 57.20 | 84.78 |
| 41 | 8 | 2 | 1 | 1 | 90 | 60 | 52.60 | 76.70 | 142.50 |
| 42 | 40 | 2 | 1 | 1 | 110 | 60 | 66.00 | 80.06 | 136.51 |
| 43 | 44 | 0 | 1 | 1 | 100 | 56 | 64.60 | 94.80 | 142.85 |
| 44 | 24 | 0 | 1 | 1 | 100 | 56 | 44.76 | 84.39 | 140.22 |
| 45 | 34 | 0 | 1 | 1 | 100 | 56 | 54.68 | 89.60 | 141.37 |

**Table A2.** Results of the ANOVA analysis of cutting force $F_x$.

| Source | DF | Adj SS | Adj MS | F-Value | *p*-Value |
|---|---|---|---|---|---|
| Model | 6 | 1133.64 | 188.941 | 2.93 | 0.019 |
| Linear | 3 | 580.91 | 193.638 | 3.00 | 0.042 |
| $x_1$ | 1 | 0.00 | 0.001 | 0.00 | 0.997 |
| $x_2$ | 1 | 37.70 | 37.700 | 0.58 | 0.449 |
| $x_3$ | 1 | 543.21 | 543.211 | 8.42 | 0.006 |
| Square | 3 | 552.73 | 184.244 | 2.85 | 0.050 |
| $x_1*x_1$ | 1 | 62.22 | 62.218 | 0.96 | 0.332 |
| $x_2*x_2$ | 1 | 59.71 | 59.706 | 0.93 | 0.342 |
| $x_3*x_3$ | 1 | 415.56 | 415.558 | 6.44 | 0.015 |
| Error | 38 | 2452.38 | 64.536 | | |
| Lack-of-Fit | 6 | 185.94 | 30.989 | 0.44 | 0.848 |
| Pure Error | 32 | 2266.45 | 70.827 | | |
| Total | 44 | 3586.03 | | | |

* represents the quadratic effect.

**Table A3.** Results of the ANOVA analysis of cutting force $F_y$.

| Source | DF | Adj SS | Adj MS | F-Value | *p*-Value |
|---|---|---|---|---|---|
| Model | 6 | 12,556.1 | 2092.7 | 8.47 | 0.000 |
| Linear | 3 | 2569.2 | 856.4 | 3.47 | 0.025 |
| $x_1$ | 1 | 1748.0 | 1748.0 | 7.07 | 0.011 |
| $x_2$ | 1 | 352.4 | 352.4 | 1.43 | 0.240 |
| $x_3$ | 1 | 468.9 | 468.9 | 1.90 | 0.176 |
| Square | 3 | 9986.9 | 3329.0 | 13.47 | 0.000 |
| $x_1*x_1$ | 1 | 3557.1 | 3557.1 | 14.39 | 0.001 |
| $x_2*x_2$ | 1 | 1682.2 | 1682.2 | 6.81 | 0.013 |
| $x_3*x_3$ | 1 | 4069.9 | 4069.9 | 16.47 | 0.000 |
| Error | 38 | 9391.8 | 247.2 | | |
| Lack-of-Fit | 6 | 2198.1 | 366.4 | 1.63 | 0.171 |
| Pure Error | 32 | 7193.7 | 224.8 | | |
| Total | 44 | 21,947.9 | | | |

* represents the quadratic effect.

**Table A4.** Results of the ANOVA analysis of cutting force $F_z$.

| Source | DF | Adj SS | Adj MS | F-Value | *p*-Value |
|---|---|---|---|---|---|
| Model | 6 | 12,281.2 | 2046.9 | 6.91 | 0.000 |
| Linear | 3 | 7035.7 | 2345.2 | 7.91 | 0.000 |
| $x_1$ | 1 | 1684.4 | 1684.4 | 5.68 | 0.022 |
| $x_2$ | 1 | 379.6 | 379.6 | 1.28 | 0.265 |
| $x_3$ | 1 | 4971.7 | 4971.7 | 16.78 | 0.000 |
| Square | 3 | 5245.4 | 1748.5 | 5.90 | 0.002 |
| $x_1*x_1$ | 1 | 3453.8 | 3453.8 | 11.66 | 0.002 |
| $x_2*x_2$ | 1 | 298.5 | 298.5 | 1.01 | 0.322 |
| $x_3*x_3$ | 1 | 1094.3 | 1094.3 | 3.69 | 0.062 |
| Error | 38 | 11,259.9 | 296.3 | | |
| Lack-of-Fit | 6 | 2278.0 | 379.7 | 1.35 | 0.263 |
| Pure Error | 32 | 8981.9 | 280.7 | | |
| Total | 44 | 23,541.1 | | | |

* represents the quadratic effect.

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
