# Peer review of "Improvement in the Hard Milling of AISI D2 Steel under the MQCL Condition Using Emulsion-Dispersed MoS2 Nanosheets"

_lubricants, doi:10.3390/lubricants8060062_

Round 1
Reviewer 1 Report
Please highlight the novelty of this work.
Discuss the mechanisms happening at primary and secondary deformation zones in the light of results obtained. The connection is lacking in manuscript.
Author Response
RESPONSE TO THE REVIEWER 1
We are very grateful for the reviews provided by the editors and each of the external reviewers of this manuscript. Please see below, our detailed response to comments.
- Please highlight the novelty of this work.
Answer:
The author would like to say thank you very much for the reviewer’s comments. The novelty of this work is summarized as below:
In this study, MoS2 nanoparticles are produced by exfoliation of liquid N2–quenched bulk by using our own method rather than buying from others. The produced MoS2 nanosheets are directly used for MQCL hard milling process.
From the obtained results, the proper nanoparticle concentration and cutting speed can be chosen to achieve the technological and economic characteristics.
The machining performance of hard milling process improves under MQCL condition by using MQCL device named by Frigid-X Sub-Zero Vortex Tool Cooling Mist System with real cooling effect based on the principle of Ranque-Hilsch vortex tube. The cooling and lubricating effects significantly enhance due to the use of novel MQCL device and MoS2 nanofluid with excellent lubricating property. Therefore, the normal coated cemented carbide inserts like APMT 1604 PDTR LT30 could be effectively used for hard machining of AISI D2 (52-60 HRC) with high cutting speed of 110 m/min, which is about 2.0-2.2 times higher than the manufacturer’s recommendations [40]. This will contribute to enlarge the cutting applicability and decrease the manufacturing cost.
Also, this is one of the first studies of this type of MQCL tool using nanofluid for difficult-to-cut materials. Compared to our previous study, which concentrated on the surface roughness, surface microstructure, and surface profile and used the commercial MoS2 nanoparticles [19], this work is a big development because MoS2 nanoparticles are produced by our own method and cutting forces Fx, Fy, Fz are investigated in detail, from which the technical guides are provided and further work can be made.
Furthermore, this MQCL tool only needs ordinary air rather than cryogenic CO2 or N2 for cooling, which will simplify the equipment and reduce the cost.
- Discuss the mechanisms happening at primary and secondary deformation zones in the light of results obtained. The connection is lacking in manuscript.
Answer:
One of the characteristic phenomena occurring in the machining of hardened steels with geometrically defined cutting tools is the formation of saw-tooth chips. Catastrophic failure causing saw-tooth chip formation in the primary shear zone is usually attributed to either cyclic crack initiation and propagation or to the occurrence of a thermoplastic instability (See in Ref.2). Thermal stresses originate mainly in the intensive friction between flank wear land and the workpiece. Due to machining hardened materials, the plastic deformation is very little, so the thermal stresses and intensive friction cause white layer (See Ref. 2). Therefore, the presence of MoS2 nanosheet having low friction coefficient and excellent lubricating property in MQCL based fluid contributes to reduce the friction stresses in contact zones, from which the cutting forces decrease significantly.
The explanation above is added to the revised manuscript.
Reviewer 2 Report
Please revised the manuscript according to comments given in the attachment. Refers to highlighted in yellow colours.
Why the second-order model was used for the force components relationships?
Why was the model developed for different force components? Can the force components be combined to represent one machining force and thus, only one model can be described?
What are the R2 of the generated models?
Please explain the scientific reasons for most of your findings. For example, why 1.0% nano-concentration lead to the lowest cutting force?
Why 110 m/min contributed to the lowest cutting force?
Author Response
RESPONSE TO THE REVIEWER 2
We are very grateful for the reviews provided by the editors and each of the external reviewers of this manuscript. Please see below, our detailed response to comments.
- Please revised the manuscript according to comments given in the attachment. Refers to highlighted in yellow colours.
Answer:
Because of the problem with opening the file uploaded in the website, I had to ask the Assistant Editor to provide the manuscript file again. However, the authors did not see the comments of the reviewer highlighted in yellow colours.
- Why the second-order model was used for the force components relationships?
Answer:
For RSM design, a second order model is generally used to approximate the response once it is realized that the experiment is close to the optimum response region where a first order model is no longer adequate. The second order model is usually sufficient for the optimum region, as third order and higher effects are seldom important.
- Why was the model developed for different force components? Can the force components be combined to represent one machining force and thus, only one model can be described?
Answer:
The authors would like to say thank you for the reviewer’s suggestion. We developed differrent force components because we want to study in detail.In detail, Fy is the thrust force, the so-called springback of the work materials, which plays a significant role in hard machining and has a strong effect dimensional accuracy and flank wear. Moreover, it causes high contact stresses over the tool flank contact face and this is a distinguishing feature of hard machining. Fz is the main cutting force, which has a strong effect on the cutting performance and productivity (please see Ref. [2]). This explanation is added to the discussion of revised manuscript.
- What are the R2of the generated models?
Answer:
Here are the R2 of the generated models of cutting forces Fx, Fy, Fz obtained from Minitab 18 software. They are already added to the revised manuscript.
For Fx
Model Summary
|
S |
R-sq |
R-sq(adj) |
R-sq(pred) |
|
8.03346 |
89.96% |
85.56% |
77.68% |
For Fy
Model Summary
|
S |
R-sq |
R-sq(adj) |
R-sq(pred) |
|
15.7211 |
94.21% |
92.45% |
88.58% |
For Fz
Model Summary
|
S |
R-sq |
R-sq(adj) |
R-sq(pred) |
|
17.2137 |
98.17% |
97.62% |
95.24% |
- Please explain the scientific reasons for most of your findings. For example, why 1.0% nano-concentration lead to the lowest cutting force?
Answer:
Based on our studies and other works on the use of MoS2 nanofluid, it reveals that the nano-concentration of MoS2 nanoparticles in the based fluid is a very sensitive factor. If we don’t use the proper nano concentration, the machining responses will be negative. In the previous studies (please see Ref. 19), we deeply investigate surface roughness, the novel observation is that the so-called “micro bubbles” remain on the machined surface shown in Figure 2, which is totally different to the case of using pure fluid (Figure 1). The morphology of MoS2 nanoparticles is ellipsoidal and they possess the large surface area; therefore, they remain on the machined surface and form a thin protective film, which amplifies when the nanoparticle concentration increases (please see Ref.41) and contributes to form the tribofilm easily (please see Ref.42). In addition, the oil mist containing MoS2 nanoparticles plays an important role in improving the cooling and lubricating characteristics in cutting zone. However, when the concentration of MoS2 nanoparticles in emulsion-based fluid rises to 0.8 wt %, it causes a negative effect on surface roughness. On the other hand, cutting forces tend to reduce when we increase the nano concentration, which means more nanoparticles presenting in oil mist to directly spray to cutting zone, so they contribute to decrease the friction coefficient.
In Ref. 41, the authors found out that a thin protective film forms, amplifies and disappears along with the rise of MoS2 nano concentration, so 1.5% nano concentration is not proper for forming the thin protective film. However, we have to make more studies to prove and will discuss in the next paper. We highly appreciate the reviewer’s comment.
|
(a) |
(b) |
Figure 1. Microstructure (a) and profile (b) of machined surface under MQCL condition with pure emulsion-based fluid. (Please see Ref. 19)
|
(a) |
(b) |
Figure 2. Microstructure (a) and profile (b) of machined surface under MQCL condition with emulsion-based nanofluid of MoS2 0.5wt%. (Please see Ref. 19)
- Why 110 m/min contributed to the lowest cutting force?
Answer:
We chose the range of investigated cutting speed, which was based on our previous studies. From the obtained results, the relation between the cutting speed and cutting force reflects the growing trend from 90 m/min to 100 m/min, and the cutting forces reduce when rising from 100 m/min to 110 m/min. This observation also fits with the other studies [2] (please see the following image). The cutting forces will continuously reduce with the increase of cutting speed. However, the cutting speed could not be risen to higher levels because we want to ensure the proper tool life. Furthermore, this range of cutting speed is much higher than that of manufacturer’s recommendations.
Reviewer 3 Report
It is recommended that authors refer to plasma cutting, high-pressure water cutting and electro-drilling.
The purpose of the study should be stated in the abstract and Introduction.
It is necessary to add a few sentences explaining the exfoliation method - because the article is also for laymen.
What do the authors understand by the term technical, economical and environmental characteristics? - it rather needs to be specified.
Is it possible to ensure the dimensional stability of the object as a result of such machining - are there residual surface stresses and how to counteract them?
Has been agglomeration of MoS2 nanoparticles observed during processing?
In: ‘2.1. The Production of MoS2 Nanoparticles’ – it is needed to more accurately provide pore distribution (standard deviation).
Figure 1 – it is needed to enter the exact Raman shift peak values ​​in Figure.
Was the contact temperature or near the contact temperature monitored during processing?
Were humidity and dustiness controlled?
Figure 5 - which means 'C, AA, A, CC, B, BB' - it is better to also put explanations in the figure caption.
Could the effect of hardness and concentration of MoS2 nanoparticles on cutting forces be compared to the results characteristic of similar machining? - this is rather a recommendation.
Style of References need correction
Author Response
RESPONSE TO THE REVIEWER 3
We are very grateful for the reviews provided by the editors and each of the external reviewers of this manuscript. Please see below, our detailed responses to comments.
- It is recommended that authors refer to plasma cutting, high-pressure water cutting and electro-drilling.
Answer:
The authors would like to say thank you for your suggestion of non-traditional cutting processes. In order to machine difficult-to-cut materials, many approaches have been proposed and studied, which include grinding and hard machining processes by using geometrically defined cutting tools, such as hard turning, hard milling, and hard drilling, and non-traditional cutting processes. In this study, the authors aim to study MQCL using nanofluid assisted hard milling process in order to improve the machining performance.
The purpose of the study should be stated in the abstract and Introduction.
Answer:
The purpose of the study highlighted in red color is stated in the abstract and introduction in the revised manuscript following the reviewer’s comment.
It is necessary to add a few sentences explaining the exfoliation method - because the article is also for laymen.
Answer:
Revised accordingly:
The temperature dramatically dropped from 80oC to -196oC when the MoS2 solution was immersed in liquid N2. This prompt temperature decreasing not only expand the MoS2 but also allow the intercalation of both K+ ions and N2 gas into the interlayer resulting in exfoliating MoS2 nanosheets under the sonication synergistic.
- What do the authors understand by the term technical, economical and environmental characteristics? - it rather needs to be specified.
Answer:
The term technical, economical and environmental characteristics made from this work can be specified as below:
The application of MQCL using MoS2 nanofluid for hard milling process improves the cutting performance. The normal carbide inserts like APMT 1604 PDTR LT30 could be effectively used for machining hardened steel AISI D2, grouped in difficult-to-cut materials while remaining good surface quality and tool life. Moreover, the cutting condition is much larger than that of manufacturer’s recommendations, so the productivity improves significantly. In addition to that, the use of very small amount of cutting fluid combined with very low nano concentration still brings out the big improvement.
- Is it possible to ensure the dimensional stability of the object as a result of such machining - are there residual surface stresses and how to counteract them?
Answer:
For machining soft materials, the plastic deformation is large, so the surface hardening and residual surface stresses are also large. On the other hand, the plastic deformation is very little in machining hard materials so that it has a very small effect on the residual surface stress. Thermal stresses originate mainly in the intensive friction between flank wear land and the workpiece, which mainly cause white layer (See Ref. 2)
- Has been agglomeration of MoS2 nanoparticles observed during processing?
Answer:
The agglomeration of MoS2 nanoparticles in emulsion cutting fluid will occur rapidly when we stop the ultrasonic pulses. To avoid this problem, the nanofluid was placed in the ultrasonicator and directly used for MQCL system as shown in the following figure.
Figure. The nanofluid placed in the ultrasonicator and directly used for MQCL system
- In: ‘2.1. The Production of MoS2 Nanoparticles’ – it is needed to more accurately provide pore distribution (standard deviation).
Answer:
We provide the lateral size statistics which estimate from the SEM data in figure 1(c).
Revised accordingly:
“Figure 1 displays the transmission electron microscopy (TEM) image, scanning electron microscopy (SEM) image, lateral size distribution and Raman spectra of MoS2 flake which obtained from N2-quenched exfoliation method [37]. A transparent to the electron beam corresponding to sheet-like structures of MoS2 flakes with a lateral size of approximate several hundreds nanometer was shown in figure 1(a) and the thickness ranges from 10 to 20 nm. Figure 1(b) displays the SEM image of MoS2 flakes which clearly show the sheets morphology with the lateral size from several hundreds nm to approximate 2 µm. Furthermore, a statistics was estimated from 75 objects in SEM images to evaluate the lateral size distribution of MoS2 nanosheets as shown in figure 1(c). Almost MoS2 nanosheets reveal the lateral size smaller than 1µm which is consistent with the TEM data. In addition, Raman spectra in figure 1(b) reveals a typical of in-plane E12g and out-plane A1g Raman peaks of MoS2 flakes located at 376 and 403.2 cm-1, respectively, which highly agreed with previous report [38-39].”
- Figure 1 – it is needed to enter the exact Raman shift peak values ​​in Figure.
Answer:
The Raman peak was indexed as shown in figure 1(d)
- Was the contact temperature or near the contact temperature monitored during processing?
Answer:
The author would like to say thank you very much for the reviewer’s suggestion. The study of cutting temperature is an important and interesting issue, which is needed to investigate for hard machining, especilly for that using nanofluid. The authors are setting up the devices and constructing the simulation models to study. This will be discuss in the next paper.
- Were humidity and dustiness controlled?
Answer:
It is a valuable comment. We also encountered with this problem when doing the experiments. In our workshop, we already had the exhaust fans and ventilation system to remove the oil mist.
- Figure 5 - which means 'C, AA, A, CC, B, BB' - it is better to also put explanations in the figure caption.
Answer:
The explanations are added in the figure caption following the reviwer’s comment.
- Could the effect of hardness and concentration of MoS2 nanoparticles on cutting forces be compared to the results characteristic of similar machining? - this is rather a recommendation.
Answer:
It is a valuable comment. Our group also studied those effect on hard turning and hard drilling. Please see Refs. [17], [31]. More studies will be made to understand the machining mechanism deeper.
[17] Tran Minh Duc; Tran The Long. Tran Quyet Chien (2019). Performance Evaluation of MQL Parameters Using Al2O3 and MoS2 Nanofluids in Hard Turning 90CrSi Steel. Lubricants, 7 (5), 1-17. Doi: 10.3390/lubricants7050040.
[31] Duc, T. M., Long, T. T., & Van Thanh, D. (2020). Evaluation of minimum quantity lubrication and minimum quantity cooling lubrication performance in hard drilling of Hardox 500 steel using Al2O3 nanofluid. Advances in Mechanical Engineering 12(2) 1–12. Doi: 10.1177/1687814019888404.
- Style of References need correction
Answer:
Thank you. The authors will revise the style of References following the journal format.
Round 2
Reviewer 1 Report
The publication can be accepted.